# Prenatal Screening and Diagnostic Considerations for 22q11.2 Microdeletions

**DOI:** 10.3390/genes14010160

**Published:** 2023-01-06

**Authors:** Natalie Blagowidow, Beata Nowakowska, Erica Schindewolf, Francesca Romana Grati, Carolina Putotto, Jeroen Breckpot, Ann Swillen, Terrence Blaine Crowley, Joanne C. Y. Loo, Lauren A. Lairson, Sólveig Óskarsdóttir, Erik Boot, Sixto Garcia-Minaur, Maria Cristina Digilio, Bruno Marino, Beverly Coleman, Julie S. Moldenhauer, Anne S. Bassett, Donna M. McDonald-McGinn

**Affiliations:** 1Harvey Institute for Human Genetics, Greater Baltimore Medical Center, Baltimore, MD 21204, USA; 2Cytogenetic Laboratory, Department of Medical Genetics, Institute of Mother and Child, Kasprzaka 17a, 01-211 Warsaw, Poland; 3Center for Fetal Diagnosis and Treatment and the 22q and You Center, The Children’s Hospital of Philadelphia, Philadelphia, PA 19104, USA; 4R&D Department, Menarini Biomarkers Singapore, Via Giuseppe di Vittorio 21/b3, 40013 Castel Maggiore, Italy; 5Department of Maternal Infantile and Urological Sciences, Sapienza University of Rome (Italy), Viale del Policlinico 155, 00161 Roma, Italy; 6Center for Human Genetics, Herestraat 49, 3000 Leuven, Belgium; 7Division of Human Genetics, The 22q and You Center, and Clinical Genetics Center, Children’s Hospital of Philadelphia, Philadelphia, PA 19104, USA; 8The Dalglish Family 22q Clinic, University Health Network, Toronto, ON M5G 2C4, Canada; 9Department of Paediatrics, Institute of Clinical Sciences, Sahlgrenska Academy, University of Gothenburg, 405 30 Gothenburg, Sweden; 10Department of Paediatrics, Queen Silva Children’s Hospital, 416 50 Gothenburg, Sweden; 11Advisium’s Heeren Loo, Berkenweg 11, 3818 LA Amersfoort, The Netherlands; 12Department of Psychiatry and Neuropsychology, Maastricht University, 6211 LK Maastricht, The Netherlands; 13Institute of Medical and Molecular Genetics, Hospital Universitario La Paz, 28046 Madrid, Spain; 14Division of Human Genetics, Ospedale Pediatrico Bambino Gesù, IRCCS, 00163 Roma, Italy; 15Department of Radiology, Children’s Hospital of Philadelphia, Philadelphia, PA 19104, USA; 16Department of Obstetrics, Gynecology, and Surgery, Perelman School of Medicine of the University of Pennsylvania, Philadelphia, PA 19104, USA; 17Clinical Genetics Research Program and Campbell Family Mental Health Research Institute, Centre for Addiction and Mental Health, and Department of Psychiatry, University of Toronto, Toronto, ON M5S 2S1, Canada; 18Division of Cardiology, Department of Medicine, and Centre for Mental Health, and Toronto General Hospital Research Institute, University Health Network, Toronto, ON M5G 2N2, Canada; 19Department of Pediatrics, Perelman School of Medicine of the University of Pennsylvania, Philadelphia, PA 19104, USA; 20Department of Human Biology and Medical Genetics, Sapienza University, 00185 Roma, Italy

**Keywords:** prenatal ultrasound, 22q11.2 deletion syndrome, noninvasive prenatal screening, preimplantation genetic testing, fetal cardiac anomaly

## Abstract

Diagnosis of a chromosome 22q11.2 microdeletion and its associated deletion syndrome (22q11.2DS) is optimally made early. We reviewed the available literature to provide contemporary guidance and recommendations related to the prenatal period. Indications for prenatal diagnostic testing include a parent or child with the 22q11.2 microdeletion or suggestive prenatal screening results. Definitive diagnosis by genetic testing of chorionic villi or amniocytes using a chromosomal microarray will detect clinically relevant microdeletions. Screening options include noninvasive prenatal screening (NIPS) and imaging. The potential benefits and limitations of each screening method should be clearly conveyed. NIPS, a genetic option available from 10 weeks gestational age, has a 70–83% detection rate and a 40–50% PPV for most associated 22q11.2 microdeletions. Prenatal imaging, usually by ultrasound, can detect several physical features associated with 22q11.2DS. Findings vary, related to detection methods, gestational age, and relative specificity. Conotruncal cardiac anomalies are more strongly associated than skeletal, urinary tract, or other congenital anomalies such as thymic hypoplasia or cavum septi pellucidi dilatation. Among others, intrauterine growth restriction and polyhydramnios are additional associated, prenatally detectable signs. Preconception genetic counselling should be offered to males and females with 22q11.2DS, as there is a 50% risk of transmission in each pregnancy. A previous history of a de novo 22q11.2 microdeletion conveys a low risk of recurrence. Prenatal genetic counselling includes an offer of screening or diagnostic testing and discussion of results. The goal is to facilitate optimal perinatal care.

## 1. Introduction

The 22q11.2 deletion syndrome (22q11.2DS) is the most common microdeletion syndrome in humans [1] and is associated with congenital anomalies and later emerging health issues. 22q11.2DS has a reported contemporary live birth incidence of 1:2148 [2] and even higher prenatal prevalence estimates of 1:992 unselected pregnancies (those without identified cardiac anomalies) and 1:1497 miscarriages [3,4], regardless of maternal age. In recent studies, a 22q11.2 deletion was found in 1:19 pregnancies undergoing diagnostic testing for cardiac anomalies and 1:93 for all indications [5]. However, many children and adults with 22q11.2DS do not have congenital heart disease (CHD) or other anomalies that would be readily detectable on routine fetal ultrasonography [6,7,8]. Despite widely available genetic testing for nearly 30 years, 22q11.2DS remains under-recognized, in part due to its multi-system nature and variable severity of associated features [1,9,10]. Many physical findings are evident at birth or soon thereafter, including CHD, especially conotruncal anomalies, and multiple other differences (Table 1).

Most children with 22q11.2DS experience delays across several developmental areas such as motor, learning/cognition, and speech and language; a minority (~30%) may have a comorbid neurodevelopmental disorder (NDD) such as mild–moderate intellectual disability (ID), attention deficit hyperactivity disorder (ADHD), and/or autism spectrum disorder (ASD) [12,13,14].

Associated features that may arise later in child to adult years include, but are not limited to, hypogammaglobulinemia, humoral deficits, atopy, autoimmune disease, failure to thrive, GERD, significant constipation, thrombocytopenia, thyroid disease (hypothyroidism in ~25%, more than hyperthyroidism), type 2 diabetes and other endocrinopathies, short stature in ~20% usually without growth hormone deficiency, skeletal issues may include cervical spine anomalies, scoliosis requiring surgery in a minority, patellar dislocation, and winged scapula. Idiopathic leg pain, fatigue, and sleep disorders ± obstructive sleep apnea may occur [15,16,17]. Treatable psychiatric illnesses, especially anxiety disorders, are common, including psychotic illness, specifically a one in four risk of developing schizophrenia [15,16,17]. Neurological conditions include idiopathic and provoked seizures with epilepsy in ~15% [10] and movement disorders including dystonia and early onset Parkinson’s disease [15,16,17]. In addition to the medical problems, many affected individuals face psychosocial issues that may impact their well-being, as well as that of their families [17].

About 85–90% of 22q11.2 microdeletions arise as de novo events, that, in contrast to trisomies, are not related to parental age. The 22q11.2 region of the human genome facilitates the potential for non-allelic unequal homologous crossover (recombination) during meiosis, which can result in microdeletions (Figure 1).

The most common 22q11.2 microdeletion, found in 85–90% of affected individuals [10], involves 2.5 to 3 megabases (Mb), extending from LCR22A to LCR22D, with resultant loss of ~90 genes [1,15,18] (Figure 2). These include *TBX1*, implicated in development of the heart, thymus, and parathyroid, and many other protein-coding genes important in organ development and function throughout life [1]. 

Smaller nested deletions overlapping the LCR22A to LCR22B and LCR22A to LCR22C (~10%) [10] regions have a similar expression. Rarer 22q11.2 deletions that overlap the more distal LCR22B to LCR22D (~5%) [10] region have phenotypes and characteristics in common with the prevalent 2.5 Mb microdeletion. Further studies are required to better understand the contribution of *CRKL* (an important cardiac and renal developmental driver) and of its neighboring genes to the phenotypic features associated with LCR22B-LCR22D or LCR22C-LCR22D, as well as the full LCR22A to LCR22D deletions [1,19,20,21]. Individuals with LCR22B-LCR22D and LCR22C-LCR22D deletions will be missed by FISH studies, as the current FISH probes (N25 and TUPLE) are located within the LCR22A-LCR22B region.

Prenatal diagnosis of 22q11.2DS is of benefit for expectant parents and affected pregnancies [1,16,17]. In addition to allowing for psychological and psychosocial preparation of the family [16,17,19], fully informed decisions regarding management of the pregnancy can be made. If a complex CHD or other relevant congenital anomaly has been detected, delivery plans can be formulated with the appropriate level of support for mother and baby. Transfer to a tertiary care center with a higher-level neonatal intensive care unit may be recommended [9,10]. Treatable issues of the affected newborn such as immune dysfunction and hypocalcemia can be anticipated and promptly addressed. Moreover, unexpected structural anomalies, such as a laryngeal web, can be identified early and corrected when needed. Importantly, a diagnostic odyssey [9,10] can be avoided, allowing for timely coordinated care from a multidisciplinary team, with potential for reducing morbidity and mortality [3]. International clinical practice recommendations are available for 22q11.2DS, for children and adults [14,17]. The latter include recommendations for prenatal and perinatal care in 22q11.2DS and genetic counselling issues [17]. Notably, universal prenatal molecular screening methods (maternal serum screening and ultrasound) do not directly assess for a 22q11.2 deletion. This review will substantially update and extend previous recommendations, providing a comprehensive overview of this important topic for contemporary practice in obstetrics, medical genetics, and related areas.

## 2. Methods

### 2.1. Literature Search

Under the auspices of the 22q11.2 Society, the international scientific organization studying chromosome 22q11.2 differences, a systematic search of the existing literature 1992–2021 was performed [14,17] that included some literature on the prenatal screening and prenatal testing options for the chromosome 22q11.2 microdeletion and 22q11.2DS, placed in a “reproductive issues” database. For the current prenatal focus, the MEDLINE database was more specifically searched on 1 October 2022, using the PubMed interface and the search terms “[(22q) OR (velocardiofacial OR velocardio-facial OR velo-cardiofacial OR velo-cardio-facial OR VCFS) OR (DiGeorge syndrome) OR (conotruncal anomaly face syndrome)] AND [prenatal]”. All identified records were screened for relevance on title and abstract. No year or language restrictions were applied. Of those reports that were deemed relevant, full-text articles were assessed for eligibility. References from articles that met inclusion criteria were catalogued and further reviewed for relevance.

### 2.2. Inclusion and Exclusion Criteria

Inclusion criteria comprised any report with relevance to prenatal issues, including prenatal screening and diagnosis of a typical 22q11.2 deletion (i.e., incorporating the LCR22A-LCR22B region at minimum but not extending beyond LCR22D). Topics included noninvasive screening with prenatal ultrasound examinations, noninvasive prenatal screen (NIPS) methods based on the analysis of cell-free DNA in the maternal plasma, and diagnostic testing in pregnancy through chorionic villus sampling and amniocentesis. Literature on reproductive options (including preimplantation genetic testing for structural rearrangements or donor gametes), and on adults with 22q11.2DS in general, were also reviewed for publications and information relevant to prenatal issues and the 22q11.2 microdeletion. No additional exclusion criteria were applied.

### 2.3. Data Extraction and Synthesis

A qualitative synthesis of the available studies was outlined by N.B. The resulting manuscript was then circulated among multiple clinicians/basic scientists from Europe, the United States, and Canada who are experienced in the clinical care of and/or laboratory testing surrounding 22q11.2DS in the prenatal/pediatric/adult settings, for additional feedback and input. The resulting consensus recommendations were further shaped by the collective group of authors with scientific evidence when possible (Figure 3).

In this rapidly evolving field, we have emphasized currently available methods, with some historic context provided as we are mindful that not all jurisdictions, or all patients, would have access to the latest technologies or associated expertise. In addition, we have provided a brief overall review of recent developments in cell-based noninvasive prenatal screening for microdeletions, given that all individuals, male and female, are at risk for de novo occurrence of the 22q11.2 microdeletion at gametogenesis (Figure 1).

## 3. Results

A total of 209 potentially relevant articles were identified in the October 2022 literature search. Screening by scanning titles and abstracts resulted in the exclusion of 109 articles, leaving 100 publications deemed relevant. Full-text screening confirmed that all 100 articles met inclusion criteria. An additional 16 papers that also met inclusion criteria were identified in a “reproductive genetics” database, a category created as part of the 22q11.2 Society initiative to update the 22q11.2DS pediatric and adult healthcare guidelines [14,17]. Thus, 116 papers formed the backbone of this review.

### Review and Recommendations 

#### 3.1.1. Diagnostic Testing for 22q11.2 Microdeletion in Pregnancy

For couples with no history of a chromosome 22q11.2 microdeletion diagnosis in either parent, there are several possible routes to prenatal diagnostic testing that could identify a fetus with a 22q11.2 deletion. Standard prenatal screening using ultrasound that demonstrates an abnormality, such as an increased nuchal translucency in the first trimester or a structural anomaly identified in the second trimester, could lead to an offer of diagnostic testing. In recent years, many pregnant families are opting to have noninvasive prenatal screening, which may include the 22q11.2 microdeletion and thus the offer of diagnostic testing in the presence of an abnormal screen result. Moreover, diagnostic testing may be performed for other indications (e.g., advanced maternal age ≥ 35 years at delivery} or personal decision).

For prospective parents (male or female) known to have a 22q11.2 deletion and thus at 50% risk of transmission, or those with a rare structural balanced chromosomal rearrangement involving the 22q11.2 deletion region, an offer of prenatal diagnostic testing would be a standard part of genetic counseling [17]. Of note, many affected parents would not be diagnosed with 22q11.2DS given under-recognition, especially in adults [17]. Some previously undiagnosed affected women will come to attention via abnormal NIPS for 22q11.2DS, or affected parents may be identified in pregnancy via a careful history eliciting associated features, e.g., palatal abnormalities, scoliosis, and cognitive deficits.

The recommended prenatal diagnostic genetic test is chromosomal microarray analysis (CMA), usually a single nucleotide polymorphism (SNP) array or oligo ± SNP array. A less expensive alternative is multiplex ligation-dependent probe amplification (MLPA). These are preferred options because they will detect all pathogenic 22q11.2-region microdeletions including those where FISH probes would miss a nested LCR22B-LCR22D or LCR22C-LCR22D deletion. However, MLPA would only be employed when there was a high suspicion of 22q11.2DS. Otherwise, CMA would be preferred, as this test will identify genome-wide copy number variants (CNVs), i.e., clinically relevant deletions and duplications that are sometimes present in addition to the 22q11.2 deletion [17]. Fluorescence in situ hybridization (FISH), a targeted test using a commercial probe (N25 or TUPLE) mapping to the LCR22A to LCR22B region, is acceptable if there is a known familial deletion that overlaps this proximal region. However, approximately 15–20% of 22q11.2 deletions will not be detected by this targeted method (Figure 2). 

For rare cases where there are typical prenatal features of 22q11.2DS but results are negative for CMA or MLPA, a panel of genes or exome sequencing are commercially available options, specifically considering pathogenic variants in genes *TBX1* (22q11.2 deletion region) and *CHD7* (at 8q12.2, associated with CHARGE syndrome) [22].

With respect to the timing of obtaining a sample for definitive diagnostic testing, this is between 10 and 13 weeks gestational age for chorionic villus sampling (CVS) and after 15 weeks gestational age for amniocentesis. These definitive methods to make a fetal diagnosis are invasive tests. A meta-analysis of studies using those that had appropriate controls found that the procedure-related pregnancy loss rates were low: 1:455 (0.22%) for CVS and 1:900 (0.11%) for amniocentesis, although the authors cautioned that the data were from experienced centers and thus may not necessarily be generalized to all centers [23]. In comparison, the authors reported the miscarriage rate (before 24 weeks) was 2.26% in the CVS control groups and 0.67% in the amniocentesis control groups [23]. For context, the general population frequency of early spontaneous pregnancy loss is 10% of all clinically recognized pregnancies [24].

#### 3.1.2. Noninvasive Prenatal Screening (NIPS) for 22q11.2 Microdeletions

NIPS has been commercially available since 2011 and has become a standard option for prenatal screening for aneuploidy. Screening for selected microdeletions, including the 22q11.2 microdeletion, has been available since 2015. The American College of Medical Genetics and Genomics has recently suggested that NIPS for 22q11.2DS be offered to all patients as a conditional recommendation, based on moderate certainty of evidence [25].

Analysis is performed on the fetal fraction of cell-free DNA (cfDNA) in the maternal plasma. Most circulating cfDNA is maternal in origin, but a minor proportion (usually ranging from 4 to 20%) is fetal, derived from the cytotrophoblasts of the placenta. NIPS is performed on a maternal blood sample after 9–10 weeks gestational age.

Recently, Dar et al. [26] reported the detection of 22q11.2 microdeletions using a single-nucleotide polymorphism (SNP)-based NIPS methodology in a prospective study of almost 21,000 women. A definitive genetic outcome was determined for ~90% of participants (n = 18,290) through prenatal diagnostic testing and/or infant samples, allowing the evaluation of sensitivity and specificity. In the sample studied, there were 12 cases of 22q11.2 microdeletion, i.e., a prevalence of 1:1542. Applying an updated NIPS algorithm, developed specifically by the authors, 10 of 12 cases were detected, with 83% sensitivity, 99% specificity, and a positive predictive value (PPV) of 52.6% (Table 2). This PPV exceeds that of the trisomy 13 general-risk population PPV of 37.2%, reported in a recent systematic evidence-based review [27]. The 10 detected cases included 4 with the typical LCR22A-LCR22D 22q11.2 microdeletion, 3 of uncertain deletion extent, and 3 proximal nested deletions. Two proximal nested deletions were not detected. Seven of the twelve cases were found in first-trimester pregnancies with none having ultrasound imaging abnormalities at that gestational age [26]. 

Previously, the SNP-based NIPS method could only detect the larger 2.5 to 3.0 Mb LCR22A-LCR22D 22q11.2 deletion, but now can identify nested LCR22A-LCR22B and LCR22A-LCR22C deletions [26]. Nested LCR22B-LCR22D and LCR22C-LCR22D deletions may be detected but this is dependent on the NIPS technology employed [19], and data are limited for these nested, less common 22q11.2 deletions.

Lin et al. [28] published the results of a retrospective study of 7826 pregnancies using a massive parallel shotgun sequencing based-NIPS to identify 13 pregnancies at high risk for a 22q11.2 microdeletion. Seven of these were confirmed by CMA for a PPV of 53.9%. The authors reported sensitivity of 100%, specificity of 99.9%, and a negative predictive value (NPV) of 98%. 

In another study using a microarray-based NIPS that targeted both the common and smaller nested 22q11.2 deletions in an enriched cohort of second-trimester pregnancies with ultrasound scan abnormalities, Bevilacqua et al. [29] reported sensitivity of 69.6% and specificity of 100%. In the general pregnant population, assuming the prevalence of 22q11.2 microdeletion to be 1 in 1000, the expected PPV is estimated to be 12.2% at 99.5% specificity and 41.1% at 99.9% specificity, while the expected NPV is estimated to be >99.9% [29]. Using the same technology in a low-risk population prospectively enrolled at a prenatal medicine department between 11 and 13 weeks (and after ultrasound evaluation), the authors reported a false positive rate of 0.27%; however, the study was not structured to calculate sensitivity or other values [30]. Despite limited familiarity with 22q11.2DS, participants reported they would recommend NIPS [29].

NIPS involves a peripheral blood sample where most DNA is of maternal origin and thus may also detect maternal CNVs, including 22q11.2 microdeletions. In one study involving retrospective analysis of 80,000 NIPS samples, there were two confirmed and one suspected maternal 22q11.2 deletions identified [31]. In another study [17,32], two maternal deletions and four maternal duplications were identified and confirmed, amongst 94,085 NIPS samples. The prevalence of maternal 22q11.2DS will be affected by study ascertainment criteria (e.g., re disadvantaged/complex populations) and by 22q11.2DS-related adult mortality and reproductive issues [16,17,33]. 

It is important to note that NIPS serves as a screening assessment but does not represent diagnostic testing for 22q11.2 microdeletions. Thus, follow-up prenatal diagnostic testing with CMA is recommended for all patients with a positive NIPS result for 22q11.2 microdeletion. When fetal CMA results are found to be normal, it is appropriate to extend CMA to maternal DNA for recurrence risk evaluation and clinical assessment and management [16,17]. 

Some patients choose to await postnatal confirmatory genetic testing. In such cases, careful fetal monitoring by ultrasound and echocardiography is recommended. Likewise, consideration of delivery at a tertiary/quaternary care facility equipped to support the potential neonate with 22q11.2DS, with all the previously discussed associated features, several of which are not identifiable by fetal imaging, should be considered where possible [14]. Parental 22q11.2DS may affect the pregnancy, delivery, and perinatal care risk assessment [17,32].

Importantly, current NIPS methods do not detect all fetal 22q11.2 deletions [26]; thus, a negative NIPS result in the context of one or more structural anomalies (or other detected features) does not significantly reduce the risk of 22q11.2DS; in such cases, a diagnostic test such as CVS or amniocentesis with CMA is recommended. 

Looking forward, the isolation of intact fetal cells from maternal circulation is a promising approach for NIPS [34]. Four different types of circulating fetal cells have been described: trophoblasts, fetal nucleated red blood cells (fnRBC), lymphocytes, and stem/progenitor cells [35]. Circulating extravillous trophoblasts (cEVTs) were the first fetal cell type found in the maternal blood, and since they are extensively released during the first trimester of pregnancy, they became an attractive target for cell-based noninvasive prenatal testing (CB-NIPT) [36,37]. As CB-NIPT is based on the analysis of pure intact fetal DNA from single cells, it is possible to perform a true noninvasive fetal genome profile for microdeletions and microduplications down to about 1 Mb in size [35]. Normal fetal profile results would reduce the residual risk for pathogenic microdeletions and microduplications. CB-NIPT is emerging as a clinical option, but large studies are needed to validate clinical performance. Moreover, cases with an abnormal profile indicating a CNV with CB-NIPT analysis would still require a confirmation using an invasive diagnostic testing method, given the (~1%) possibility of feto-placental mosaicism [35].

#### 3.1.3. Prenatal Screening—Ultrasound Imaging in 22q11.2 Microdeletions

Prenatal screening for features associated with 22q11.2DS can be performed by ultrasound examination of the fetus in all three trimesters [6,11,38,39]. In the first trimester, the nuchal translucency can be evaluated, and the fetus can be screened for severe structural anomalies. A more detailed fetal anatomic examination can be performed in the second trimester, allowing for the detection of a wide range of anomalies, including some that are subtle. In the third trimester, fetal growth and amniotic fluid variations can be assessed, and further evaluation of fetal anatomy can be performed. 

Patients with the most common (LCR22A-LCR22D) and with nested (LCR22A-LCR22B, LCR22A-LCR22C, LCR22B-LCR22D, LCR22C-LCR22D) 22q11.2 deletions (Figure 2) are considered together as a group, as only a few series offered information regarding fetal ultrasound findings based on the genomic extent of the 22q11.2 deletion (Figure 2). Individuals identified via FISH studies are expected to have at minimum a LCR22A-LCR22B deletion given that standard clinical 22q11.2-specific FISH probes are located within this region (Figure 2). CHD, detected in 36% (5/14) of fetuses with LCR22B-LCR22D or LCR22C-LCR22D deletions, was less frequent than in fetuses with typical (e.g., LCR22A-LCR22D) deletions [38,40,41,42].

#### 3.1.4. First-Trimester Ultrasound

The nuchal translucency (NT) is the collection of fluid under the skin at the back of the fetal neck seen in the first trimester of pregnancy. A cystic hygroma is a large septate NT. A thickened NT, including cystic hygroma, is associated with chromosomal abnormalities and genetic syndromes. With NT abnormalities, there is an increased risk for cardiac malformations, with greater risk if there is a cystic hygroma. In four retrospective prenatal studies of ultrasound findings, an increased NT was seen in a total of 38/470 (8%) 22q11.2DS pregnancies, with the frequency ranging from 5 to 19% in individual studies [6,11,38,39]. In contrast, a systematic review of CMA in fetuses with increased NT and normal karyotype found 0.66% had a 22q11.2 microdeletion [43].

Significant fetal malformations can be detected by first-trimester ultrasound examination. Noel et al. [6] noted anencephaly, omphalocele, and two pregnancies with hyperechoic kidneys in their series of 74 pregnancies with 22q11.2DS. A recent meta-analysis of 63 ultrasound studies found identification of over half of fetuses affected by major cardiac pathology in the first trimester [44]. Skeletal defects can also be identified. Endovaginal ultrasound may aid in obtaining appropriate images [45].

Systematic analysis of CNVs associated with early pregnancy loss showed 22q11.2 microdeletions to be most frequently involved [46]. Possibly, major cardiac malformations are incompatible with pregnancy progression and development [47]. A wide prenatal phenotypic spectrum of 22q11.2 deletions can be hypothesized, including lethal or very severe clinical features that may (or may not) be detectable at early and later gestational ages.

#### 3.1.5. Second-Trimester Ultrasound

The physical fetal anomalies in the majority of pregnancies that are diagnosed with 22q11.2DS are identified in the second trimester (Figure 4 and Figure 5). 

Cardiac malformations are the most common prenatal finding reported in 22q11.2DS fetuses. Series with at least 40 pregnancies have reported cardiac anomalies in 62–95% of pregnancies evaluated [6,11,38,39,41]. Conotruncal anomalies were the most common type of cardiac defect observed, identified in 62–78%. ToF was the most common heart defect seen, reported in 23–45% of patients, followed by IAA, TA, and conoventricular VSD. 

CHD represented an isolated finding (i.e., no other physical congenital anomaly detected) in just over half of the patients evaluated in three series [11,38,39]. Other ultrasound findings were less often solitary.

Cardiac malformations were the indication for diagnostic genetic testing in 59 of 80 patients with 22q11.2DS identified in previously unpublished prenatal data from a series using CMA at the Institute of Mother and Child, Warsaw, Poland, from 2014–2021 (Table 3) [48]. Of the cardiac anomalies reported, 49% (29/59) were ToF, 12% hypoplastic left heart syndrome (HLHS), and 19% a VSD (Table 3).

Hypoplasia or aplasia of the thymus is an emerging fetal sonographic finding for 22q11.2DS. The fetal thymus can be measured from 15 weeks gestational age. Although not usually assessed during an ultrasound examination, studies have shown that the fetal thymus can be identified in 95% of fetuses evaluated [49]. Chaoui et al. [50] noted that the fetal thymus was hypoplastic in cases of 22q11.2DS [50]. Subsequently, this group reported that the fetal thymus could be successfully evaluated by performing a thymic–thoracic ratio at the three-vessel-tracheal view of the heart and that thymic hypoplasia or aplasia as a sonographic marker for 22q11.2DS had a positive predictive value of 81.8%, sensitivity of 90%, and specificity of 98.5% [51]. Several other groups have reported on prenatal thymic findings in pregnancies affected by 22q11.2DS. Sivrikoz et al. [11] detected thymic hypoplasia or aplasia in 20 of 28 pregnancies. A small series found four of seven fetuses with thymic abnormalities [52]. Two other studies that included anatomical pathology data reported thymic hypoplasia/aplasia identified by prenatal ultrasound in only 3.7% and 2.7%, but on autopsy thymic abnormalities were seen in 53% and 86% of 83 and 72 cases, respectively [6,39]. A study performed by Dou et al. [52] to determine if the prenatal thymus size correlated with postnatal immunologic problems found that prenatal hypoplastic or absent thymus was correlated with postnatal low T cell, but not B cell, count.

Skeletal findings have been observed on prenatal ultrasound of 22q11.2DS in up to 25% of affected pregnancies. Talipes equinovarus is the most often reported malformation, seen in up to 20% of affected pregnancies [6,11,41]. Other previously observed skeletal abnormalities include anomalous vertebrae, pectus carinatum, short extremities, polydactyly, and syndactyly.

Craniofacial anomalies are reported in 1–21% of affected pregnancies [6,11,39,41]. These include small ears, cleft lip with or without cleft palate, hypotelorism, a bulbous nasal tip, hypoplastic nasal bone, and micrognathia [6,7,11,41,53]. Palatal abnormalities are more often identified postnatally [1].

Genitourinary (GU) malformations are reported in 8–17% of 22q11.2DS pregnancies, and include pyelectasis and hydronephrosis, unilateral or rarely bilateral renal agenesis, unilateral multicystic dysplastic kidney, and ureterocele [6,11,38,39,41,54,55]. GU malformations were noted in 7.5% (6/80) of the pregnancies in the Kowalczyk et al. series [48].

Prenatal gastrointestinal (GI) abnormalities are infrequently seen but may be severe. Diaphragmatic hernia and tracheoesophageal fistula have been reported in several series [11,41,54]. Imperforate anus has been identified. Other findings include right umbilical vein, umbilical vein varix, umbilical cord hernia, and single umbilical artery. The overall rate of GI anomalies reported prenatally in 22q11.2DS is 3–11% [6,11,41,54]. 

Anatomical central nervous system (CNS) anomalies have also been described in association with 22q11.2DS. The cavum septi pellucidi (CSP) is a midline brain structure, part of the longitudinal cerebral fissure, and a standard component of the fetal anatomic survey by ultrasound. An enlarged CSP has been associated with the common aneuploidies and with additional brain malformations. Chaoui et al. [56] found the CSP was dilated (>95th percentile width per biparietal diameter) in 67.5% of 37 22q11.2DS pregnancies evaluated between 16 and 34 weeks gestational age. Using this method of calculating dilation, Pylypjuk et al. [57] reported accurate detection in their cohort of 29 affected fetuses, with sensitivity of 79.3% and specificity of 94.7%. A dilated CSP was a less common finding in two other studies that noted the CSP: it was dilated in 2/76 [38] and prominent in 6/42 patients [41]. Recently, it was reported that in over 7000 ultrasounds of fetuses suspected to have a fetal anomaly and referred for second opinion, 14 of 21 with an enlarged CSP had 22q11.2DS [58].

The CSP, a normal fetal finding, closes in 85% of infants in the general population, with a minority remaining open into adulthood. Some studies report a greater than expected proportion with persisting open CSP, or wider CSP measurements, in 22q11.2DS [59,60]. It is unknown if the prenatal finding of a dilated CSP correlates with neuropsychiatric outcomes.

Other CNS anomalies that have been occasionally described in fetuses with 22q11.2DS include neural tube defects, enlarged cisterna magna, asymmetric or enlarged cerebral ventricles, and agenesis of the CSP [6,11,41,54]. It is important to recognize that most major neurodevelopmental and neuropsychiatric outcomes of concern to families are not detectable by prenatal ultrasound [61].

#### 3.1.6. Third-Trimester Ultrasound

Abnormal ultrasound findings are identified in fetuses with 22q11.2DS as late as the third trimester. Polyhydramnios has been noted in 16% of fetuses (81/512), with a range of 4–31%. It was the only finding in 0–8% of fetuses with 22q11.2DS and third-trimester ultrasound data [6,11,39,41], and in one study, five of eleven fetuses, including one identified by fetal MRI, were found to have thymic aplasia/hypoplasia as the only abnormality seen with polyhydramnios [8]. Most of the polyhydramnios was unexplained, although in a small number of pregnancies it was related to GI obstruction. Given the high prevalence of postnatal functional GI disorders (including dysphagia, GERD, and constipation found in 38%, 58%, and 60% of children with 22q11.2DS < 13 years of age, respectively), polyhydramnios might be considered an indirect predictor of GI/palatal abnormalities to come, as these are unable to be identified via prenatal imaging [62]. Further studies will be needed to determine if this is the case.

Low or absent amniotic fluid can also occur in association with 22q11.2DS, but as in pregnancies in general, this finding is often associated with renal anomalies or poor fetal growth [55].

Using small for gestational age (SGA), defined as <3rd percentile birthweight for gestational age, as a proxy for intrauterine growth retardation (IUGR), unselected population-based newborns with a 22q11.2 microdeletion had a sevenfold increased risk for SGA compared to nearly 30,000 newborns with no deletion [2]. Similarly, in a study of 123 adults with a 22q11.2 microdeletion, 25 (20.3%) were born SGA, by design none preterm or born to affected mothers; this was over eight times the population-based risk [63]. IUGR was reported in several prenatal case series of 22q11.2DS. Volpe et al. [54] noted 7/19 with IUGR; two of the IUGR pregnancies went on to be stillborn. Three other studies reporting IUGR in a combined 6 of 126 continuing pregnancies were closer to the population-based expectations [11,39,41].

#### 3.1.7. Prenatal Screening—MRI Imaging in 22q11.2 Microdeletion

Fetal MRI can be a helpful tool, e.g., when evaluating the fetal brain, heart, or thymus. Two studies noted its role in the evaluation of the fetal thymus [8,64]. Fetal MRI has been found to be effective in the assessment of fetal cardiac anatomy, particularly for a more accurate study of the arch vessels [65]. Availability of fetal MRI is limited however, and its use is generally reserved for diagnostic clarification, particularly of the brain, in centers with experience reading these images.

#### 3.1.8. Reproductive Options for a Parent with 22q11.2 Microdeletion

Issues relevant to reproductive health for prospective parents with 22q11.2DS include the availability of informed genetic counselling, effects of delayed diagnosis of 22q11.2DS, potential impact of maternal morbidities, and adequacy of social support [16,17,66]. 

Reproductive options that may be discussed with a parent with a 22q11.2 microdeletion include having no prenatal genetic evaluation, preimplantation genetic testing for structural rearrangements (PGT-SR), diagnostic genetic testing in pregnancy by CVS or amniocentesis, and noninvasive screening through ultrasound. NIPS is a screening option for a paternal 22q11.2 deletion; NIPS cannot reliably identify an affected fetus in the presence of a maternal 22q11.2 deletion. Parents should be aware of their options if a pregnancy is found to be positive for 22q11.2DS, including continuing the pregnancy, termination of pregnancy, and adoption. Besseau-Ayasse et al. [39] reported in a French study that families with a 22q11.2DS parent were as likely to terminate a pregnancy as families where the 22q11.2 deletion was found to be of de novo origin. Other reports suggest limited use of prenatal diagnostic options for parents with 22q11.2DS [33,66]. Preconception counseling is optimal [16,17,66]. 

PGT-SR involves invasive procedures. First, in vitro fertilization (IVF) is performed, and then PGT-SR is typically carried out on embryonic biopsies taken at the blastocyst trophoectoderm stage, 5 days post fertilization. Testing methodology includes FISH (with the limitations as discussed above) or whole-genome amplification-based technology, such as aCGH or SNP microarray [67]. There are two case reports of IVF/PGT-SR performed specifically for familial 22q11.2DS, the first in 1998 [68,69]. In both cases, the mother was found to have 22q11.2DS after having an affected child. Of note, the current options available for preimplantation genetic aneuploidy screening (PGT-A) are unable to detect small microdeletions or microduplications, such as the 22q11.2 microdeletion. It is thus necessary to perform specific PGT-SR when testing for 22q11.2DS. 

There are several challenges in choosing PGT-SR. It is impossible to predict the severity of the specific phenotype. The cost of IVF and of the genetic testing may be substantial; however, this cost may be weighed against that of treating an individual for a lifetime of associated features. Of note, a successful pregnancy is not guaranteed, and the entire process will often place a high psychological burden on the couple.

Other reproductive options are available, including donor gametes. If the father has a 22q11.2 deletion, artificial insemination with donor sperm can be performed. If the mother has a 22q11.2 deletion, IVF with donor eggs is an option. 

Couples who have had a child with a de novo 22q11.2 deletion have a slightly elevated risk compared to population expectations, due to the possibility of germline or gonadal mosaicism [70,71]. From a practical standpoint, since the possibility of germline mosaicism cannot be definitively ruled out, diagnostic genetic testing by CVS or amniocentesis can be offered to couples who wish to avoid this risk in a future pregnancy. Any individual seeking genetic testing for an unrelated reason, such as advanced maternal age (≥35 years at delivery), would also have this option.

Current genetic counselling should also include a discussion of marked phenotypic variability, also within affected families, that concerns every single associated manifestation of 22q11.2DS. For example, cognitive deficits appear to be greater in offspring of parents with 22q11.2DS than in offspring with de novo 22q11.2 deletions [72,73], where parental IQ appears on average to modify the significant effects of the 22q11.2 deletion itself [74]. Moreover, while there are no significant parent-of-origin differences with de novo 22q11.2 deletions [73], parent of origin may have an impact on intellectual outcomes in inherited 22q11.2 deletions. Median full-scale IQ was reported in a recent study to be significantly lower (on average by eight points) in offspring of an affected mother than an affected father [73]. There are many potential explicatory factors, including assortative mating, possible effects of maternal comorbidities, and/or that the relatively few affected men with 22q11.2DS who reproduce may have less severe neurodevelopmental phenotypes [72]. For context, the 22q11.2 deletion in general lowers IQ on average by about 30 IQ points [74]. Caution is required in providing individual risks and predicted outcomes. 

## 4. Discussion

Many individuals with a chromosome 22q11.2 microdeletion do not have congenital organ or other anomalies that would be readily detectable on routine fetal ultrasonography, especially those considered significant enough to prompt chorionic villus sampling or amniocentesis [62]. Moreover, as nearly 90% of newly diagnosed 22q11.2 microdeletions occur as a de novo event to an unaffected couple, regardless of parental age, most affected pregnancies will be found in individuals of younger maternal age, compared to those of advanced maternal age (≥35 years at delivery) when invasive prenatal genetic testing has been more often performed. Many adults with 22q11.2DS remain undiagnosed and thus do not receive preconception counselling, nor the offer of prenatal testing or other reproductive options [16,17,66]. 

CHD, especially conotruncal defects, identified by prenatal ultrasound will raise suspicion for 22q11.2DS. With improved technology, these can be diagnosed earlier in pregnancy. The prenatal phenotype has expanded, and findings such as hypoplastic thymus or dilated CSP can also represent prenatal markers of 22q11.2DS. Developing an understanding of the variable fetal presentation of 22q11.2DS, particularly in the absence of typical cardiac findings, will allow parents access to appropriate testing and thus detection of pathogenic 22q11.2 deletions. The thymus is not yet routinely evaluated as part of a detailed anatomic ultrasound or as part of a fetal echocardiogram. Implementation of this evaluation in the future, particularly when there is increased index of suspicion, could improve detection of 22q11.2 deletions. 

Progress has also been made in the prenatal genetic screening and diagnosis of 22q11.2DS (Table 4). NIPS has become a standard prenatal screen, with more patients electing to screen for CNVs, including 22q11.2 deletions. Diagnostic testing with aCGH or MLPA is being offered more often for pregnancies suspected to have 22q11.2DS and for those having diagnostic testing for other indications. Other screening options based on the analysis of isolated circulating extravillous trophoblasts from maternal blood are on the horizon and may offer an additional noninvasive screening modality to detect fetal pathogenic microdeletions and microduplications in the future.

Deferring genetic diagnostic testing to the newborn period (e.g., day of life 1) is an option for those who decline definitive invasive testing in pregnancy. Early diagnosis allows for more appropriate perinatal management and ensures neonatal support at birth according to individual needs [14,17]. For those children without prenatal features of 22q11.2DS or a positive NIPS result, universal newborn screening, such as that developed in 2012 using qPCR, as well as newer methodologies, remains the promise of tomorrow to identify all children with 22q11.2DS as early as possible and thus improve potential for anticipatory care and optimized outcomes [9,75,76]. This would include those with critical CHD and the many burdened with a protracted diagnostic odyssey [9,75,76]. 

The 22q11.2DS is quite common [2] and has been reported in first cousins by chance alone [77], as well as in combination with a secondary diagnosis including those caused by additional copy number variants, as well as familial and de novo single gene disorders [78]. Thus, when a 22q11.2 microdeletion has been confirmed in a fetus or child with an atypical presentation, consideration should be given to performing additional genetic testing, e.g., whole-exome sequencing [77,78].

Although here we report the many advances made in prenatal identification of chromosome 22q11.2 microdeletions, this work was limited by the heterogeneity of studies reviewed. Many had small sample sizes, were retrospective, and were published over the past several decades. Conversely, during this time span, ultrasound technology has improved dramatically, NIPS technologies and algorithms now provide better detection rates, and genetic testing is more comprehensive. Nonetheless, drawing overall conclusions remains premature. 

Improved screening will result in more pregnancies diagnosed with 22q11.2DS and in a minority of cases could lead to the identification of a parent who also has 22q11.2DS. In all cases, benefits will include genetic counseling and opportunities for individualized management and anticipatory care [14,17]. Challenges include the variability of the phenotype and health issues that arise from before birth to adulthood, as for many genetic conditions. Families will require updated information, guidance, and support throughout this lifetime journey [14,17]. 

## Figures and Tables

**Figure 1 genes-14-00160-f001:**
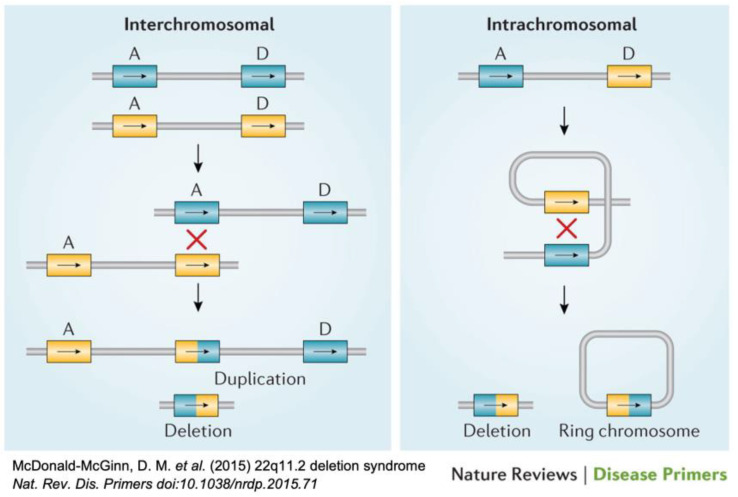
22q11.2 non-allelic homologous recombination [1]. Diagram of two different types of meiotic non-allelic homologous recombination events that can occur between low copy repeats on chromosome 22 (LCR22s), at spermatogenesis or oogenesis. Rearrangements between LCR22A and LCR22D are indicated (A and D) on each allele (blue versus yellow). Interchromosomal events (**left**) occur between paralogous LCR22s (A and D) in two different alleles owing to >99% sequence identity of direct repeats (“X” shows the crossover of the two chromosomes). The hybrid LCR22 is shown as half yellow and half blue. This process results in a duplication or deletion of the intervening region and genes in resulting gametes. Intrachromosomal recombination events (**right**) result from crossing over (indicated by red “X”) within one allele, resulting in a deletion (**left**) or a ring chromosome (**right**); the ring chromosome is not viable [1].

**Figure 2 genes-14-00160-f002:**
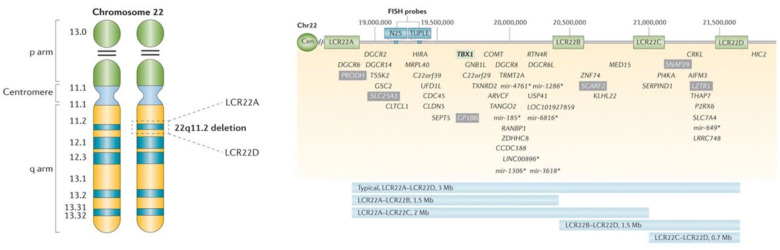
Chromosome 22 ideogram and low copy repeats that span this region. Chromosome 22 ideogram (**left**). Cytogenetic representation of chromosome 22 showing the short (p) and long (q) arms along with the centromere, which functions to separate both arms. Chromosome 22 is an acrocentric chromosome, as indicated by the two horizontal lines in the p arm. The 22q11.2 deletion occurs on the long arm of one of the two chromosomes, depicted by dashed lines in the 22q11.2 band. The position of the two low copy repeats (LCRs) on 22q11.2 (LCR22A and LCR22D), which flank the typical 2.5 to 3 Mb deletion (length varies when including/not including the LCRs), are indicated. Low copy repeats and genes within the 22q11.2 deletion (**right**). Schematic representation of the 2.5 to 3 Mb 22q11.2 region that is commonly deleted in 22q11.2 deletion syndrome, including the four low copy repeats (LCR22s) that span this region (LCR22A, LCR22B, LCR22C, and LCR22D). Common commercial probes for fluorescence in situ hybridization (FISH) are indicated (N25 and TUPLE). Selected protein-coding (and non-coding*) genes are indicated with respect to their relative position along chromosome 22 (Chr22). T-box 1 (*TBX1*; green box) is highlighted as the most widely studied gene within the 22q11.2 region. Mutations in this gene have resulted in conotruncal cardiac anomalies in animal models and humans. The CRK Like Proto-Oncogene, Adaptor Protein, also known as V-Crk Avian Sarcoma Virus CT10 Oncogene Homolog-Like, (*CRKL*; located between LCR22C and LCR22D) gene has been associated with renal/urogenital and congenital cardiac anomalies [1].

**Figure 3 genes-14-00160-f003:**
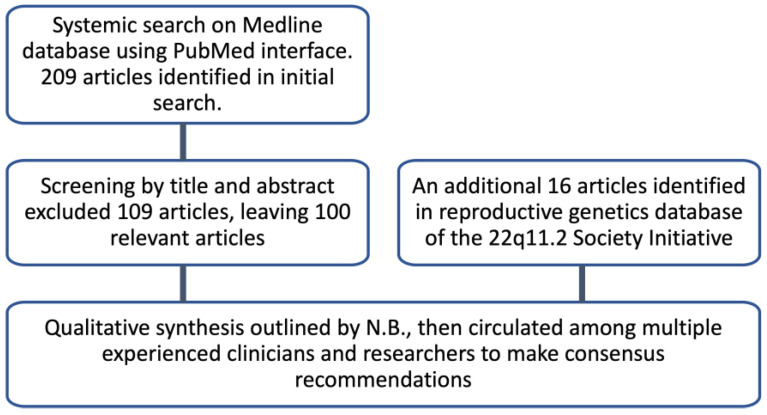
Flow chart of search strategy methods and results.

**Figure 4 genes-14-00160-f004:**
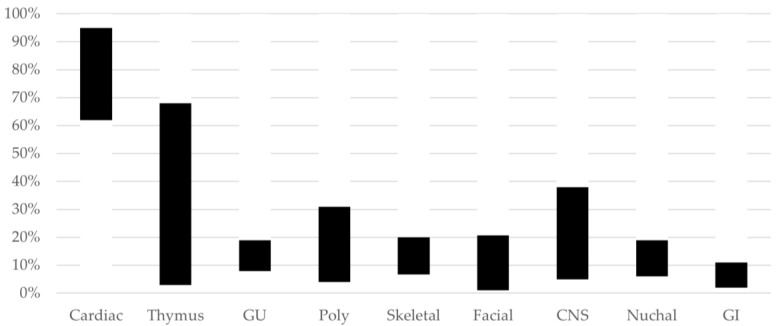
22q11.2DS Structural Findings. Range of proportion of fetuses with prenatal structural findings in nine categories, later found to have a diagnosis of 22q11.2DS. Based on data from studies with 40 or more patients [6,11,38,39,41]. Legend: GU, genito-urinary tract; Poly: polyhydramnios; CNS: central nervous system, GI: gastrointestinal.

**Figure 5 genes-14-00160-f005:**
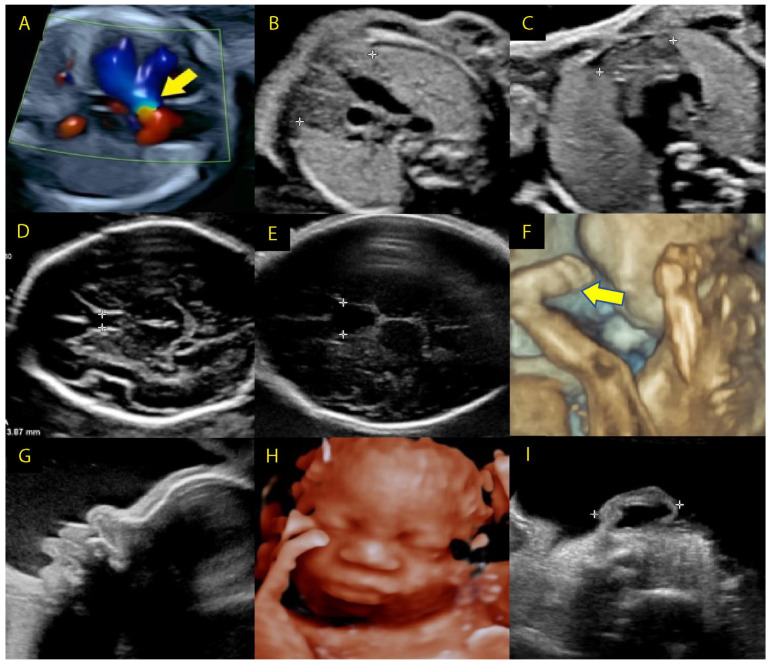
Second- and third-trimester ultrasound findings in 22q11.2DS. (**A**) Tetralogy of Fallot, overriding aorta (arrow), (**B**) normal thymus, (**C**) hypoplastic thymus, (**D**) normal cavum septi pellucidi (CSP), (**E**) enlarged CSP, (**F**) talipes equinovarus (arrow), (**G**) profile with bulbous nose, (**H**) face with bulbous nose, (**I**) small ear with thickened overfolded helix. (**B** through **E**—areas of interest are indicated by calipers).

**Table 1 genes-14-00160-t001:** Features of 22q11.2 deletions that may be evident prenatally, at birth, or soon thereafter.

System Affected	Examples
Congenital heart disease (CHD) (64%) [10] (conotruncal anomalies, aortic arch anomalies)	Tetralogy of Fallot (ToF) (18%) [10] Pulmonary atresia ± ventricular septal defect (PA/VSD) (6%) [10] Truncus arteriosus (TA) (4%) [10] Interrupted aortic arch (especially type B) (IAA) (symptomatic once the ductus arteriosus has spontaneously closed, which may be post neonatal discharge) (11%) [10] Conoventricular septal defects (VSD)/muscular VSD (23%) [10] Aortic arch and epiaortic vessel anomalies (AAA), including double aortic arch or right aortic arch (RAA), with/without aberrant subclavian arteries (ASC) ± resulting in a double aortic arch/vascular ring (14%) [10]
Palatal abnormalities (67%) [10]	Velopharyngeal insufficiency (most frequently manifesting in infancy as nasal regurgitation) (52%) [10] Submucosal cleft palate ± bifid uvula (21%) [10] Overt cleft palate (6%) [10] Cleft lip and palate (1–2%) [1]
Genitourinary anomalies	Renal anomalies (16%) [10] Bilateral or unilateral renal agenesis Multicystic dysplastic kidneys Hyperechoic kidneys Vesicoureteral reflux Inguinal hernia Hypospadias (4%) [10] Cryptorchidism (4%) [10]
Skeletal differences	Scoliosis (50%—most frequent onset in adolescence) [10] Cervical spine anomalies (46%—unlikely to be observed prenatally) [10] Talipes [11] Butterfly vertebrae [11] Preaxial and postaxial polydactyly of the hands [1,10] Postaxial polydactyly of the feet [1,10]
Immunodeficiency-related	Thymic aplasia or hypoplasia [11] T cell lymphopenia (50%) [10]
Endocrinopathies (>50%) [10]	Neonatal hypocalcemia due to hypoparathyroidism ± with neonatal seizures (55%) [10]
Otolaryngologic anomalies	Laryngeal web/subglottic stenosis (21%) [10] Trachea-esophageal fistula Esophageal atresia, tracheal atresia
Gastrointestinal problems (65%) [10]	Feeding and swallowing difficulties (30%) [10] Umbilical hernia Omphalocele [11] Imperforate anus [1,10] Intestinal malrotation or non-rotation [1,10] Hirschsprung’s disease [1,10] Congenital diaphragmatic hernia (CDH) [1,10]
Neurologic manifestations	Hypotonia [1,10] Idiopathic seizures (15%) [10] Neonatal seizures/jitteriness/cyanosis (may be due to hypocalcemia) [10] Microcephaly [1] Polymicrogyria, heterotopias [10] Open operculum [1] Chiari malformation [10] Tethered cord [1] Neural tube defects: myelomeningocele, anencephaly [10,11]

**Table 2 genes-14-00160-t002:** 22q11.2 microdeletion detection by NIPS.

Reference	Sensitivity%	Specificity%	Positive Predictive Value (PPV)%	Negative Predictive Value (NPV)%
Dar et al., 2022 [26]	83.3	99.8	52.6	99.9
Lin et al., 2021 [28]	100	99.9	53.9	99.9
Bevilacqua et al., 2021 [29]	69.6 *	100 *	100 *	98 *

Design, sample acquisition, methods, and ability to detect nested 22q11.2 deletions differ for each study (see text for details). * Enriched cohort with ultrasound scan abnormalities of CHD and 6.3% prevalence of 22q11.2DS, and thus high PPV.

**Table 3 genes-14-00160-t003:** Prenatal diagnostic findings: n = 80 identified to have a 22q11.2 deletion at Institute of Mother and Child, Warsaw, Poland.

Presenting Prenatal Finding	Total	Isolated	Multiple	Additional Findings
**Cardiac**	**59**	**45**	**14**	
ToF	29	25	4	Enlarged nuchal, VSD, AAA + ARSA, Pyelectasis, Hydronephrosis
AAA	2	1	1	Hydronephrosis + megaureter
ARSA	2	-	2	Cardiac outflow tracts, VSD
IAA	3	3	-	-
Truncus arteriosus	2	1	1	VSD
Truncus arteriosus communis	2	-	2	VSD, RAA
Coarctation of aorta	1	-	1	VSD
HLHS	7	6	1	Omphalocele
VSD	11	9	2	Micrognathia, Polydactyly
**Other ultrasound findings**	**9**	**7**	**2**	
Enlarged nuchal	2	2	-	-
Urinary tract	3	3	-	Renal diastases, Megacystic
CDH	1	1	-	-
Cerebral ventriculomegaly	1	-	1	Hernia + family history
Talipes equinovarus	1	1	-	-
Multiple anomalies	1	-	1	Not specified
**Abnormal screening**	**11**	**10**	**1**	
Serum screening elevated for trisomy	10	9	1	Umbilical hernia
NIPS high risk 22q11.2 deletion	1	1	-	-
**Parent with 22q11.2 deletion**	**2**	**2**	**-**	

Aberrant right subclavian artery (ARSA), coarctation of aorta (COA), congenital diaphragmatic hernia (CDH), epiaortic vessels anomalies/aortic arch anomaly (AAA), interrupted aortic arch (IAA), hypoplastic left heart syndrome (HLHS), noninvasive prenatal screening (NIPS), right aortic arch (RAA), tetralogy of Fallot (TOF), truncus arteriosus (TA), ventriculoseptal defect (VSD).

**Table 4 genes-14-00160-t004:** Prenatal guidelines for the 22q11.2 microdeletion.

**Genetic diagnosis of 22q11.2 microdeletion in conceptus/embryo/fetus**
Prenatal testing using chromosomal microarray or MLPA to analyze chorionic villus sample (10–13 weeks gestational age) or chromosome preparations from fetal cells obtained by amniocentesis (beginning at 15 weeks gestational age). FISH is an option when there is a known familial 22q11.2 deletion involving the typical deletion region. Preimplantation genetic testing (PGT-SR) of a fertilized embryo, by IVF, is available in many locations.
**Genetic screening of 22q11.2 microdeletion in embryo/fetus**
**Noninvasive prenatal genetic screen (NIPS)** NIPS can be offered from 9 weeks gestational age; sensitivity and specificity of the method used need to be discussed. **Ultrasound examination for potential fetal anomalies** First trimester for increased nuchal translucency, severe structural anomalies. Second-trimester high-resolution ultrasound examination, including screening for cardiac, renal, skeletal, and other anomalies (best from 18 weeks gestation); assess thymus when possible. Fetal echocardiogram (18–22 weeks gestational age). Third trimester for polyhydramnios, intrauterine growth restriction (IUGR), further assessment for anomalies.
**When fetus has tested positive for a 22q11.2 microdeletion**
Genetic counseling regarding prognosis, including that prenatal ultrasound examination does not rule out all anomalies and does not predict postnatal course or lifetime features of 22q11.2DS. Clinical genetic diagnostic testing of both parents to determine whether one has the 22q11.2 deletion, expected in 10–15% of cases. Consider delivery at a tertiary care center with knowledge and resources to care for neonates with 22q11.2DS.
**When a parent has a 22q11.2 microdeletion**
Provide genetic counseling, ideally preconception, including options as above.

## Data Availability

Not applicable.

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
