# Peer review of "Prenatal Screening and Diagnostic Considerations for 22q11.2 Microdeletions"

_genes, 2023, doi:10.3390/genes14010160_

Round 1

Reviewer 1 Report

In the Article “Prenatal Screening and Diagnostic Considerations for 22q11.2 Microdeletions”, Blagowidow and colleagues provide a review and assessment of screening (both genetic and imaging) and diagnostic options for this commonly encountered microdeletion syndrome.

The manuscript is well written by a team who appear to have extensive experience with this patient group. It is detailed and adds value to the literature on this subject. There are some areas that could be further clarified, including:

1.     Table 1:  would be enhanced by the % of time that finding is encountered, either the category of defect, or a specific defect, similar to what is seen in Smith’s for many disorders. For example, there was work published in NEJM several years ago, noting that renal malformations are found in 40% of affected patients.

2.     Similarly, is it known how often autism disorders are seen in patients with 22q deletions?

3.     In the paragraph following Table 1, there are other entities that are encountered but they are all co-mingled. Could they go either into another table of less frequent findings or at least separated into categories?

4.     On page 4, in the lone paragraph that describes “nested” deletions, is this the same labeling used for proximal, central and distal deletions? [Rump et al, AJMG 2014; Ben Schachar, AJHG, 2008]

5.     Most NIPS tests and arrays describe a 2.54 Mb deletion, where does that fit in the 3 Mb deletion shown in Figure 2?

6.     Figure 2 appears to be from a Nature Reviews article from 2015, are the genes and gene names in the figure still the same?

7.     What is the purpose of Figure 3? Is it necessary?

8.     Can the term “micro-imbalance” on p. 7 be replaced with either microdeletion/duplication or copy number variant? This sounds like a small unbalanced translocation.

9.     In section 3.1.8 which discusses options for parents who are themselves affected, it states that NIPS can be used. Can all NIPS tests be used or only certain ones? Will a mother’s deletion interfere with the testing across any of the available NIPS tests?

10.  In the cited paper by Dar, the PPV increased after the use of an AI-based algorithm and one more case was detected. Do all of the commercial NIPS use this approach and have similar results?

11.   ACOG and SMFM do not currently recommend using NIPS for 22qdel because of the issues with PPV, do the authors think this should be changed? Also, can the authors provide any insight on why the PPV continue to be so low for this deletion syndrome as opposed to others?

Author Response

Reviewer 1:

Points 1-3 – As suggested, we have revised Table 1 and the manuscript text in the Introduction (p 3, lines 83-96) to provide further information about approximate prevalence of features of 22q11.2DS. We have been conservative, as there are insufficient data available, and substantial variability in expression depending on ascertainment, age/developmental stage, etc., to be more precise with respect to percentages. We have however maintained a system-based structure to present features of this multi-system genetic condition.

Points 4-6 – With respect to the issue of “nested” 22q11.2 deletions, we have further clarified in the Introduction (p 4, lines117-121) that we are referring to the four main recurrent nested 22q11.2 deletions that occur flanked by the low copy repeats (LCRs) A, B, C, and D, all of which are within the common LCR22A-LCR22D 22q11.2 deletion region. We have explicitly stated that the review does not include the “distal” 22q11.2 deletions, i.e., distal to LCR22D. These points are further emphasized by a revised Figure Legend for Figure 2 and with additional emphasis on the gene that has been most studied within the deletion (TBX1) and the other main cardiac and renal developmental gene (CRKL) in the text.

The common 22q11.2 (LCR22A-LCR22D) deletion region is usually reported as 2.5 Mb to 3 Mb in length, depending on whether the flanking LCRs are included or not. We have now clarified this in the figure legend for Figure 2 and have ensured that we report both lengths throughout the manuscript.

Point 7 – Figure 3 is, we believe necessary for this review to summarize the method used.

Point 8 – As requested, we have replaced the term, “micro-imbalance” with “microdeletion” (p 7, line 201).

Point 9 – The reviewer correctly points out an important limitation of NIPS, and we have now clarified in the text in Section 3.1.8 (p 14, lines 524-526) that NIPS is an option for paternal screening, but not reliable for maternal 22q11.2DS

Point 10 – Regarding the algorithm issue for NIPS methods, (p 8, line 278), it is unique; this is now noted in the paper.

Point 11 – We have now further addressed the issue of PPV for NIPS detection of 22q11.2 microdeletions (p 8, lines 264-267). ACMG has just given a conditional recommendation supporting NIPS for 22q11.2DS. ACOG/SMFM often follow with a similar recommendation in 1-2 years. The PPV for 22q11.2DS is currently superior to that for trisomy 13 (in the general population) in most labs and we have noted that as well (p 8, lines 280-281). Data for other microdeletion NIPS is limited, in part due to the lower incidence of these other microdeletions.

Reviewer 2 Report

This is a very necessary and outstanding review of prenatal detection of 22q11.2 deletions and duplications.  It is fully comprehensive of all available published knowledge on the matter and very clearly written. Prenatal detection of such a frequent de novo genomic error is very important for parents and clinicians.  For this reason, the field is in great need of a review on this topic.

Minor points: 

Although the manuscript reads very well, the use of Introduction, Methods, Results and Discussion headings are somewhat ackward for a review.  Authors may consider restructuring the manuscript with other headings that are more informative of the content for the different sections or do away with them.

The first two figures that are reproduced from Nature Reviews disease primers have really large lettering with the Nature logo.  I am not sure this is necessary for a figure in another journal from another editor.

Finally, in line 332 the sentence is somewhat confusing.  It states:

"NIPS results (or other detected features) does not significantly reduce the risk of the condition"

I believe what the authors mean is that it does not reduce the risk of the condition in the parents.

If not, please clarify.

Author Response

Reviewer 2:

Point 1 – With respect to the headings used for this Review, we have followed the Journal standard, so have not changed these. We have however made some clarifications in wording in each section that will hopefully assist the reader in this regard.

Point 2 – Regarding Figures 1 and 2 reproduced from the 2015 Nature Reviews Primer paper, we have followed the reviewer’s suggestion and have modified these with respect to the Nature logo.

Point 3 – Further to the Reviewer’s suggestion, we have clarified the wording regarding negative NIPS results not significantly reducing risk of a 22q11.2 microdeletion (p 9, lines 365-369).
